# Molecular Biology of the *WWOX* Gene That Spans Chromosomal Fragile Site *FRA16D*

**DOI:** 10.3390/cells10071637

**Published:** 2021-06-29

**Authors:** Cheng Shoou Lee, Amanda Choo, Sonia Dayan, Robert I. Richards, Louise V. O’Keefe

**Affiliations:** Department of Molecular and Biomedical Science, School of Biological Sciences, The University of Adelaide, Adelaide, SA 5000, Australia; cslee0820@gmail.com (C.S.L.); amanda.choo@adelaide.edu.au (A.C.); sonia.dayan01@gmail.com (S.D.)

**Keywords:** common chromosomal fragile sites, *FRA16D*, megabase gene, intragenic homozygous deletion, oxido-reductase specificity, evolutionary conservation

## Abstract

It is now more than 20 years since the *FRA16D* common chromosomal fragile site was characterised and the *WWOX* gene spanning this site was identified. In this time, much information has been discovered about its contribution to disease; however, the normal biological role of *WWOX* is not yet clear. Experiments leading to the identification of the *WWOX* gene are recounted, revealing enigmatic relationships between the fragile site, its gene and the encoded protein. We also highlight research mainly using the genetically tractable model organism Drosophila melanogaster that has shed light on the integral role of *WWOX* in metabolism. In addition to this role, there are some particularly outstanding questions that remain regarding *WWOX*, its gene and its chromosomal location. This review, therefore, also aims to highlight two unanswered questions. ***Firstly***, what is the biological relationship between the *WWOX* gene and the *FRA16D* common chromosomal fragile site that is located within one of its very large introns? ***Secondly***, what is the actual substrate and product of the *WWOX* enzyme activity? It is likely that understanding the normal role of *WWOX* and its relationship to chromosomal fragility are necessary in order to understand how the perturbation of these normal roles results in disease.

## 1. Chromosomal Fragile Site Genes—The Precedent of *FRA3B/FHIT*

Chromosomal fragile sites are of interest for a number and variety of reasons [1]. They are non-staining gaps in chromosomes that can be induced to appear by specific chemicals in cell culture medium. They differ in their frequency in the population. ***Rare*** fragile sites are only found in some individuals in the population. The rare fragile sites are due to expanded DNA repeats, with those individuals expressing the fragile site having a copy number above the threshold for cytogenetic appearance. A relationship of some sort exists between the chemistry of induction and the DNA sequence composition—AT-binding/substituting chemicals have AT-rich expanded repeats. ***Common*** fragile sites can be induced to appear in everyone’s chromosomes; however, they vary in their sensitivity to induction. Inhibitors of DNA polymerase induce most common fragile sites and their appearance is, therefore, related to replication. The common fragile sites vary in the frequency with which they respond to induction—the *FRA3B* site on human chromosome 3 being most readily observed, followed by *FRA16D* on chromosome 16, then others [1].

A correlation between cancer cell DNA instability and chromosomal fragile sites had been noted long ago [2], although the notion that such a relationship was causal had been greeted with some scepticism [3,4]. A relationship of some sort received a boost of interest with the finding that the most readily observed common chromosomal fragile site, *FRA3B*, was located within a region on human chromosome 3 that exhibited DNA instability in cancer [5]. Furthermore, the *FHIT* gene was found to span the *FRA3B* common chromosomal fragile site and aberrant transcripts of the *FHIT* gene were found in cancer cells [6]. A functional role for *FHIT* as a tumour suppressor, however, also turned out to be controversial with conflicting data regarding the contribution of *FHIT* to cancer [7,8]. In one study using neoplastic cells that had *FRA3B* deletions and, therefore, were deficient in *FHIT* protein, “replacement” with stable, over-expressed *FHIT* protein did not alter in vitro or in vivo properties of these cells [9]. In another study [10], replacement of *FHIT* protein in cancer cells suppressed their tumorigenicity. To further add to the mystery, “enzyme inactive” mutant *FHIT* was just as effective as the normal active *FHIT* at suppressing tumourigenicity [10], implying that its 5′,5”-P^1^,P^3^-triphosphate hydrolase activity is not required for all of its functions. The review by Glover et al. [11] details recently reported mechanisms of the cytogenetic expression of common chromosomal fragile sites and some of the controversy around whether or not *WWOX* and *FHIT* are actually tumour suppressor genes.

## 2. Chromosomal Fragile Site *FRA16D,* Cancer and the *WWOX* Gene

From the outset, the *WWOX* gene was unusual. Indeed, the experiments leading to the identification of the *WWOX* gene are noteworthy as they reveal a number of significant (and unexpected) characteristics, which have an impact on the encoded protein and its function [12,13,14,15,16]. Some of these are yet to be explained.

A major reason for interest in this region of the genome was based upon the consequences of the Knudsen hypothesis [17]—that inherited cases of cancer were higher than sporadic cases because the latter required two somatic mutations to a tumour suppressor gene while the familial cases only needed one. One of the forms of second mutation that validated Knudsen’s hypothesis was loss-of-heterozygosity [18] and so the search for regions of the genome that exhibited loss-of-heterozygosity in cancer was thought to be a means of tracking down novel tumour suppressor genes. The *FRA16D* region had been found to be within overlapping regions of loss-of-heterozygosity in breast [19] and prostate [20] cancers, suggesting the presence of a tumour suppressor. The presence of *FRA16D,* the second most readily observed common chromosomal fragile site in the human genome, contributed to speculation of a causal relationship between DNA fragility and instability and the presence of a tumour suppressor gene. Indeed, both Mangelsdorf et al. [12] and Paige et al. [13] identified homozygous deletions within several cancer cell lines that coincided with the location of *FRA16D.*

The gene that has come to be known as *WWOX* (WW-containing Oxidoreductase, Bednarek et al. [14]) was first located in the *FRA16D* region by Paige et al. [13] as *HHCMA56*, an oxidoreductase encoding sequence that had been deposited in GenBank in 1994 by Gmerek, R.E. and Medford, J.I. Paige et al. [13] had excluded *HHCMA56* from contention on the basis of a PCR (D16S432E) that located (the final exon) of this gene hundreds of kilobases in distance from the “minimal homozygously deleted region in cancer cells” identified by Mangelsdorf et al. [12] and Paige et al. [13]. Oxidoreductases were not amongst known tumour suppressors at the time and *HHCMA56*, therefore, appeared to have unlikely credentials as such, although some members of this protein family have since been found to have modifying roles in cancer [21,22]. The *FRA16D* minimal deletion region was in due course sequenced by Ried et al. [15] (GenBank accession number AF217490) as it was expected to contain one or more exons of a tumour suppressor gene. Instead, the gene responsible for *HHMCA56* was found to be huge and indeed span *FRA16D*. With hindsight, such a possibility might have been considered given that the huge *FHIT* gene spanned the *FRA3B* fragile site.

*HHCMA56* was a partial cDNA sequence from one of a number of alternatively spliced RNA transcripts. *D16S432E* is located in its unique 3′ exon (corresponding to exon 9 of *WWOX*). Exon 8 of *WWOX* is shared between two alternatively spliced transcripts (named *FOR I* and *FOR II* by Ried et al. [15]). The common exon 8 sequences were found at the very beginning of the AF217490 sequence (indeed, prior to the minimally deleted region) and the alternative exon 9 from the *FOR I* transcript at the other end. Finnis et al. [23] subsequently found that some homozygous deletions in cancer cells are, indeed, only intronic, which appears at odds with such deletions knocking out a tumour suppressor gene. This intron is 260 kb in length in the minor *FOR I* transcript and a massive 780 kb in length in the major *FOR II* (*WWOX*) transcript. Furthermore, both transcripts share intron 5, which is also a massive 222570 bases in length (Figure 1).

The gene was subsequently named *WWOX* (WW domain containing Oxido-reductase) by Bednarek et al. [14], while Ried et al. [15] had given the gene the name of *FOR* (Fragile site *FRA16D* Oxido-Reductase). Chang et al. [16] identified the gene by virtue of its induction by hyaluronidase and named it *WOX1*. The *WWOX* gene has, therefore, accumulated multiple alternate names (*FOR, WOX1, DEE28, EIEE28, FRA16D, SCAR12, HHCMA56, PRO0128, SDR41C1,* and *D16S432E*).

The relationship between long genes and chromosome fragility is noteworthy, as the *FHIT* gene spanning *FRA3B* is also very large (at ~1.5 Mb). The orthologous mouse *Fhit* gene also spans a common chromosomal fragile site [24], as does the mouse *WWOX* gene [25]. The biological pressure during evolution to maintain chromosomal susceptibility to environmental agents within very long genes is, therefore, intriguing. Indeed, the unusual length of the *WWOX* gene has been retained through evolution—even amongst species such as *Fugu* and *Drosophila* that typically have very much shorter introns in their gene orthologues (Figure 1). Given the risk that common chromosomal fragile sites confer as target sites for DNA instability, the conservation of *WWOX* gene/intron length suggests some form of biologically advantageous relationship; however, the basis for this, and what (if any) benefit it may confer, are not yet apparent.

The *WWOX* gene is of remarkable length for a protein of only 414 amino acids. Its primary transcript is over 1.1Mb in length, of which >98% is intron. For comparison, another member of the SDR family to which *WWOX* belongs is hydroxysteroid 17-beta dehydrogenase 1, which is encoded by the gene *HSD17B1*. This protein of 328 amino acids is translated from a mature mRNA of 1274 nucleotides having been spliced from a primary gene transcript of 2292 nucleotides in length. Not only does the human *WWOX* gene have vastly larger introns that another member of the same enzyme encoding gene family, but this extreme length of introns has been conserved through evolution, even in organisms that typically have short introns, i.e., *Fugu* and *Drosophila* (Figure 1). The parallels with the *FHIT* gene that spans the *FRA3B* common chromosomal fragile site are striking and of relevance to further properties of *WWOX*, as discussed later in this review and elsewhere [26].

Steady-state protein abundance is determined by four rates: transcription, translation, mRNA decay and protein decay [27]. The ability of a gene to produce a protein product is determined by transcription, which takes time. Transcription rates vary widely; however, it is safe to assume that a primary transcript of the *WWOX* gene takes several, if not many, hours to complete and is significant in relation to the time necessary for the cell cycle in dividing cells. *FRA16D*-associated intronic deletions might be expected to hasten the process; however, introns and their splicing can enhance gene expression [28]. Adding to the enigma, *WWOX* primary transcripts undergo alternative splicing with only one form encoding the full-length protein. Typically, alternatives to the full-length transcript are subject to non-sense mediated decay and contribute to a reduction in the steady-state level of mRNA for the full-length protein. Driouch et al. [29], however, report a substantially elevated level of an alternatively spliced transcript (designated *FORIII* by Ried et al. [5]) in ~50% of breast cancer tissues and cell lines. This perturbed splicing occurs in the absence of detectable DNA deletions within the *WWOX* gene, further contributing to the enigma. Its relevance to cancer cell biology is, as yet, unknown.

The enormous length of the *WWOX* gene and its alternative splicing would appear to be two of the contributing factors to *WWOX* protein having a low steady-state level. Indeed, *Drosophila* go one step further with the presence of an intron in the 3′ untranslated region (UTR) in some *WWOX* transcripts (see *Wwox-RB* versus *Wwox-RA* transcripts in FlyBase (https://flybase.org/reports/FBtr0343384 (Date last reviewed: 15 November 2018)). Bignell et al. [30] report that such mRNAs with 3′UTR introns are subject to non-sense mediated decay after only a single round of translation, indicating another mechanism for keeping *WWOX* protein levels low. Whether human *WWOX* RNA transcripts also have such 3′UTR introns may warrant further investigation as, according to Bignell et al. [30], it is often assumed that such sequences are non-functional.

The properties of the mutation that gives rise to homozygous deletion at the *FRA16D* fragile site in cancer cells have been explored and are noteworthy [23]. First, the early timing of the deletion event in the neoplastic process, as it is assumed that early events are more likely to be causal rather than consequential. Secondly, the lack of a relationship between *FRA16D*-associated deletion and another form of deletion (loss-of-heterozygosity, LOH) known to occur at high frequency in certain cancers in the 16q23.2 region [19,20]. Thirdly, the nature of the deletion endpoints suggests a specific form of DNA deletion repair mechanism [31]. Fourthly, the extent of “genome-wide” instability that occurs in *FRA16D* deleted cell lines. Finally, the lack of impact of *FRA16D*-associated DNA deletions on the ability to cytogenetically express the *FRA16D* fragile site. 

The relationship between *FRA16D* homozygous deletion and the loss-of-heterozygosity is particularly noteworthy. Finnis et al. [23] detail the experimental basis for the conclusion that the *FRA16D*-associated homozygous deletion events observed in cancer cell lines in this manuscript are distinct from the loss-of-heterozygosity observed by others in breast [19] and prostate [20] cancers. In brief, the polymorphic genetic markers *D16S518* and *D16S504* that define the boundaries of the loss-of-heterozygosity regions identified in cancer [19,20], were found by Finnis et al. [23] to be heterozygous in all of the cancer cell lines that exhibited *FRA16D* homozygous deletion. This indicates that the homozygous deletions observed, at least in the particular cancer cells under investigation (i.e., AGS, HCT116, CO-115, KM12C and KM12SM), are not able to be the boundaries of any loss-of-heterozygosity that may have occurred in the vicinity in these cells. Whether this nexus is broken in other instances is yet to be determined. It is perhaps noteworthy that the KM12 lines had a common origin (being derived from the primary, KM12C and metastasis, KM12SM of the same cancer) and had identical homozygous deletions at *FRA16D* yet exhibited different DNA instabilities elsewhere (including a chromosomal translocation). Fragile site DNA instability is, therefore, not always tied to other instances of DNA instability that presumably have different causes.

Common chromosomal fragile sites exhibit a hierarchy of cytogenetic expression, with *FRA3B* being more readily observed than *FRA16D.* DNA instability in cancer cells at *FRA3B* is also more frequent than that at *FRA16D.* This finding contributes to a growing body of evidence suggesting that common fragile sites are regions of particular sensitivity to DNA instability and that there is a correlation between the level of in vitro chromosomal fragility and in vivo DNA instability in cancer cells [26,32,33]. The localised multiple-hit nature of the homozygous mutation, together with its subsequent (relative) stability, suggests that it is most likely that a transient interaction between environmental factors plays a determining role in the common fragile site-associated mutation mechanism.

## 3. WWOX in Metabolism

Despite more than twenty years of research on the *WWOX* protein, the substrate and product of the enzyme reaction that it catalyses are yet to be discovered. A growing body of evidence in various model systems supports a role for *WWOX* in metabolism (see [32,33] for extensive reviews). *Drosophila* deficiency in *WWOX* displays no phenotypic consequences [34] and, therefore, might be considered a poor model for those species (including humans) for which *WWOX* is necessary. On the contrary, the ability of *Drosophila* to compensate for the lack of *WWOX* indicates that pathology caused by deficiency of *WWOX* is likely to be treatable, with identification and targeting of the compensating pathway(s).

A combination of *Drosophila* genetics and biochemical approaches was utilised to discover the normal function of the *WWOX* gene [34,35,36,37]. Genetically altered levels of *WWOX* resulted in the identification by proteomics and microarray analyses of multiple components of aerobic metabolism. Functional relationships between *WWOX* and two of these, isocitrate dehydrogenase or Cu–Zn superoxide dismutase, were confirmed by genetic interactions. In addition, altered levels of *WWOX* resulted in altered levels of endogenous reactive oxygen species. Similarly to *FHIT*, *WWOX* contributes to pathways involving aerobic metabolism and oxidative stress, providing an explanation for the “non-classical tumour suppressor” behaviour of *WWOX*. Fragile sites, and the genes that span them, are therefore part of a protective response mechanism to oxidative stress and likely contributors to the differences seen in aerobic glycolysis (Warburg effect) in cancer cells [32,34]. 

In support of these findings in *Drosophila*, experiments in human HEK392T cells have demonstrated that *WWOX* has an interrelationship with metabolism—*WWOX* is both a regulator of metabolism and is regulated by metabolism [35]. Alteration of growing conditions from oxidative phosphorylation to glycolysis alters the expression levels of *WWOX*. Under hypoxic conditions where metabolism is steered towards glycolysis, the expression of *WWOX* transcript is markedly decreased, whereas a switch to oxidative phosphorylation has the opposite effect. *WWOX* not only contributes to the regulation of homeostasis, its steady-state levels are linked to the state of cellular metabolism.

An insight into the contribution *WWOX* plays in cancer was revealed by competition experiments that showed a role for *WWOX* in the elimination of tumourigenic cells [36]. *WWOX* was first shown to modify *TNF*-mediated cell death phenotypes, which was reflected in changes to Caspase 3 staining and provided evidence for *WWOX* in the promotion of cell death. These *TNF*-mediated cell death phenotypes were shown to correspond to increased levels of reactive oxygen species (ROS), which have also previously been shown to be regulated by *WWOX* [36]. Together, these data suggested a protective role for *WWOX* in the promotion of cell death in response to increased ROS levels, which could correlate with altered metabolism that is observed in cancer. Indeed, decreased levels of *WWOX* within clones of tumorigenic cells resulted in fewer of them being eliminated by the surrounding wild-type cells and worse outcomes at later stages [36]. These studies provided a molecular basis for *WWOX* acting as a suppressor of tumor growth by mediating cell death pathways. Together, these results provide a molecular basis for the non-classical tumour suppressor functions of *WWOX* and the better prognosis observed in cancer patients with higher levels of *WWOX* activity.

Furthermore, *WWOX* acts to moderate the mitochondrial respiratory system—a likely contribution to the Warburg effect [37]. An in vivo genetic study using *Drosophila melanogaster* revealed a role for *WWOX* in a mitochondrial-mediated pathway dependent on its SDR enzyme function. Reduced levels of *WWOX* were found to result in further perturbation of cellular dysfunction caused by mitochondrial deficiencies, leading to increased frequency of phenotypes such as loss of tissue, cellular outgrowths and presence of ectopic structures. Conversely, the tissue disruption phenotypes were suppressed by increasing *WWOX* levels, with the SDR enzymatic active site required for the suppression. Amino acid Y288 in *Drosophila* is an essential component of the catalytic active site in the SDR region, with Y288F mutation abolishing its function [38], and similar mutations shown to completely abolish enzymatic activity of other SDR proteins [38,39,40]. The orthologous tyrosine amino acid is position 293 in human *WWOX*. The *WWOX* proteins of different species vary in length due to the presence/absence of additional amino acids. *Drosophila* experiments utilising the Y288F mutation therefore demonstrated that the catalytic activity of *WWOX* is required for its cellular response to mitochondrial defects. These experiments indicate the participation of *WWOX*, through its SDR enzyme activity, in the maintenance of cellular homeostasis in response to mitochondrial defects. Reduction in *WWOX* levels leads to a lessened cellular response to metabolic perturbation of normal cell growth caused by mitochondrial damage-induced glycolysis (Warburg effect).

Experiments from Aqeilan et al. assign contributions of *WWOX* to various components of general metabolism [33]. *WWOX* regulates glucose metabolism via *HIF1α* modulation [41], while loss of *WWOX* activates aerobic glycolysis [42] and the somatic ablation of *WWOX* in skeletal muscles alters glucose metabolism [43]. Furthermore, the *WWOX* gene modulates high-density lipoprotein and lipid metabolism [44]. Pathway analysis of *WWOX* interactors by Lee et al. [45] identified a significant enrichment of metabolic pathways associated with proteins, carbohydrates, and lipids breakdown.

## 4. WWOX Genomic Region Is a Risk Factor in Metabolic Disorders

Metabolic dysfunction is a defining feature of chronic human diseases including Type 2 Diabetes, hypertension, heart disease, chronic obstructive pulmonary disease, obesity and numerous forms of cancer. It is clear that there are genetic risk factors that predispose individuals to such diseases and/or affect the course of disease progression. Genetic risk is indicative of roles for specific proteins and the pathways in which they are rate-limiting determinants. The genomic region containing the *WWOX* gene has been identified as a genetic risk factor in each of these metabolic diseases [46,47,48,49,50,51,52,53,54,55,56,57]. 

The maintenance of metabolic homeostasis is vital to health and its disruption is central in many of the most costly human diseases. Cellular metabolism is highly integrated with multiple mechanisms in place to monitor and restore homeostasis. A key element of intracellular metabolism is the balance between glycolysis and oxidative phosphorylation in the generation of ATP from carbohydrate. *WWOX* is both regulated by perturbation in the oxidative phosphorylation/glycolysis balance, as well as being a regulator of this balance [34,35,36,37]. This integral role of *WWOX* in homeostasis therefore provides a plausible explanation for metabolic disorders where genetic variation of *WWOX* is a risk-factor.

## 5. Outstanding Questions

### 5.1. Why Is the Chromosomal Fragile Site FRA16D Located within the WWOX Gene?

While a great deal of focus has understandably been on the contribution of *WWOX* to disease [46,47,48,49,50,51,52,53,54,55,56,57,58,59,60,61], the normal role of the *WWOX* gene and its encoded protein remain mysterious. The *WWOX* gene has not one but two massive introns, the larger of which contains the common chromosomal fragile site—a region of sensitivity to environmental agents (Figure 1). The presence of large introns is conserved through evolution. Indeed, organisms that typically have much shorter introns than mammals (e.g., *Fugu rubripes* and *Drosophila melanogaster*) also have uncharacteristically large introns in their *WWOX* genes (Figure 1). This relationship between fragile site containing genes and the very large length of their primary transcripts suggests a role for transcription timing in the regulation of *WWOX* expression. Genes of 1Mb in length will take many hours to produce a full-length transcript, raising a curious relationship to the cell cycle during replication.

The location of regions of greater environmental sensitivity within certain genes is also curious. DNA damage occurs at greater frequency at common chromosomal fragile sites and is a hallmark of cancer. The common fragile site genes are responsive to changes in metabolism and at least *WWOX* and *FHIT* contribute to homeostasis [16,34,35,36,37]. Therefore, the possibility exists that the evolutionarily conserved presence within certain genes of DNA sequences that are sensitive to DNA damage is part of a biologically advantageous response mechanism to environmental damage—not merely conferring risk to cancer.

### 5.2. What Does WWOX the Enzyme Normally Do?

Given the rate-limiting role of *WWOX* in metabolism and its high degree of conservation during evolution, it is surprising that the enzymatic reaction catalysed by *WWOX* is unknown. By sequence homology, *WWOX* protein encodes a small chain dehydrogenase (SDR) enzyme with a requisite NAD(P)[H] co-factor binding site [38,39,40]. SDR enzymes typically have small molecule substrates and catalyse the interconversion of C-O-H with C=O and the resultant generation of NAD(P)+ or NAD(P)[H]. The *WWOX* orthologues of all species show significant, and as yet unexplained, homology in their C-terminal sequences (see Figure 2). Whilst conserved during evolution, these sequences do not exhibit any detectable homology to known protein motifs. They look to be a unique property of the *WWOX* protein and, therefore, somehow related to its unique biological function.

Kavanagh et al. [62], in their review entitled “The SDR superfamily: functional and structural diversity within a family of metabolic and regulatory enzymes”, state the following: “The common mechanism is an underlying hydride and proton transfer involving the nicotinamide and typically an active site tyrosine residue, whereas substrate specificity is determined by a variable C-terminal segment”. Whatever the biological basis of the very high level of conservation during evolution in the C-terminal region of *WWOX*, it is reasonable to speculate that sequences outside of the immediate catalytic motif (and C-terminal to it) have a role to play in contributing to conformation of the catalytic site and/or the access of molecules to that catalytic site and, therefore, the specificity of the enzyme.

Of relevance, homozygous mutations that diminish *WWOX* function are found in families with recessive spinocerebellar ataxia 12 (*SCAR12*). The G372R mutation located within the putative substrate specificity region indicates that this highly conserved C-terminal segment is vital for *WWOX* function, having the same clinical consequences as that of the P47T mutation located in the first WW domain of *WWOX* [57]. Given that these conserved C-terminal sequences do contribute to *WWOX* substrate-specificity, they are potential targets in the identification of *WWOX*-based therapeutics.

The presence of two WW domains that are known to act as protein–protein interaction sites with PPY containing proteins has focussed much attention on the identity of the protein binding partners of *WWOX* in an effort to ascertain which biological pathways and processes *WWOX* contributes to [63]. In addition to the intriguing findings revealed by phylogenetic analysis of the *WWOX* gene, a similar analysis of *WWOX* protein also adds to the mystery. Most organisms have a single clear orthologue of *WWOX* with the characteristic two WW domains. However, *Caenorhabditis elegans* and Opisthokonts (*Casaspora owczarzaki*) are notable exceptions in that their closest *WWOX* orthologues are devoid of WW domains. A host of PPY containing proteins have been identified by various means; however, the extent to which these contribute to functional interactions in vivo is not yet clear [63].

Another striking feature of the comparison of *WWOX* proteins from different species is the conservation of 15 amino acids that include part of the first WW domain (Figure 2). This sequence has the defining characteristics of a PEST domain [64]. Proteins containing PEST domains are rapidly degraded and are found in proteins associated with the cell cycle [65]. PEST domains are well known regulators of enzyme activity [66] and are typically found in metabolically unstable proteins. These characteristics, along with that of an exceptionally long primary transcript described above, are consistent with a dynamic role for *WWOX* enzyme activity in metabolic processes integral to the cell cycle. These factors likely combine with alternative splicing and non-sense mediated decay of *WWOX* transcripts (as described above) to produce a low steady-state level of *WWOX* protein.

Notably, similarly to its *FRA3B/FHIT* counterpart, the role of *FRA16D/WWOX* in cancer has been controversial [16], resulting in their categorisation by some as “non-classical” tumour suppressor genes. Loss of a single allele (LOH, loss-of-heterozygosity) is typical in cancer, resulting in reduced *WWOX* levels rather than its absence due to a second-hit mutation. Therefore, the altered abundance of *WWOX* metabolites appears sufficient for biological consequences, including poorer prognosis for cancers with reduced WWOX [32,58]. Reduced *WWOX* enzyme activity suggests a build-up on one side of the equation, which leads to elevated levels of a metabolite and, therefore, has biological consequences. The identity of this metabolite, together with targeted methods to reduce its abundance, represent a plausible target for treating metabolic dysfunction due to perturbation of *WWOX*. Alternatively, if the product(s) of *WWOX* act as negative regulators or rate-limiting determinants of a metabolic process, then the activation of compensatory pathways such as those acting in *Drosophila* [34] may provide a means of reducing the clinical impact of *WWOX* deficiency.

## Figures and Tables

**Figure 1 cells-10-01637-f001:**
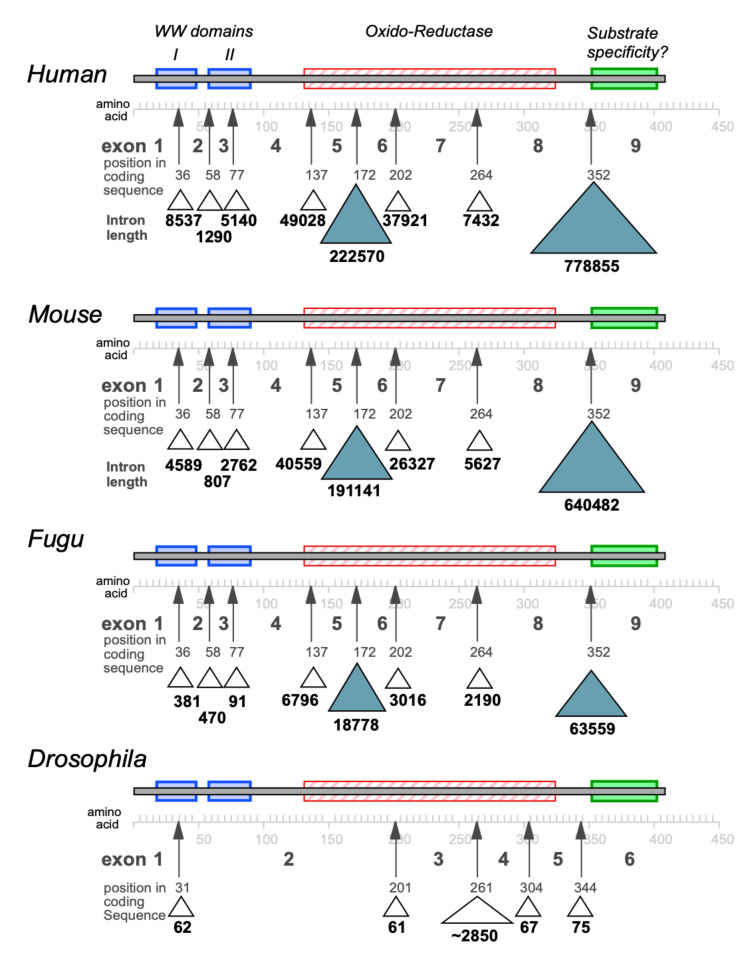
Location and length of introns in the WWOX genes of different species—with respect to major functional domains of the *WWOX* proteins of human, mouse, *fugu* and *Drosophila*.

**Figure 2 cells-10-01637-f002:**
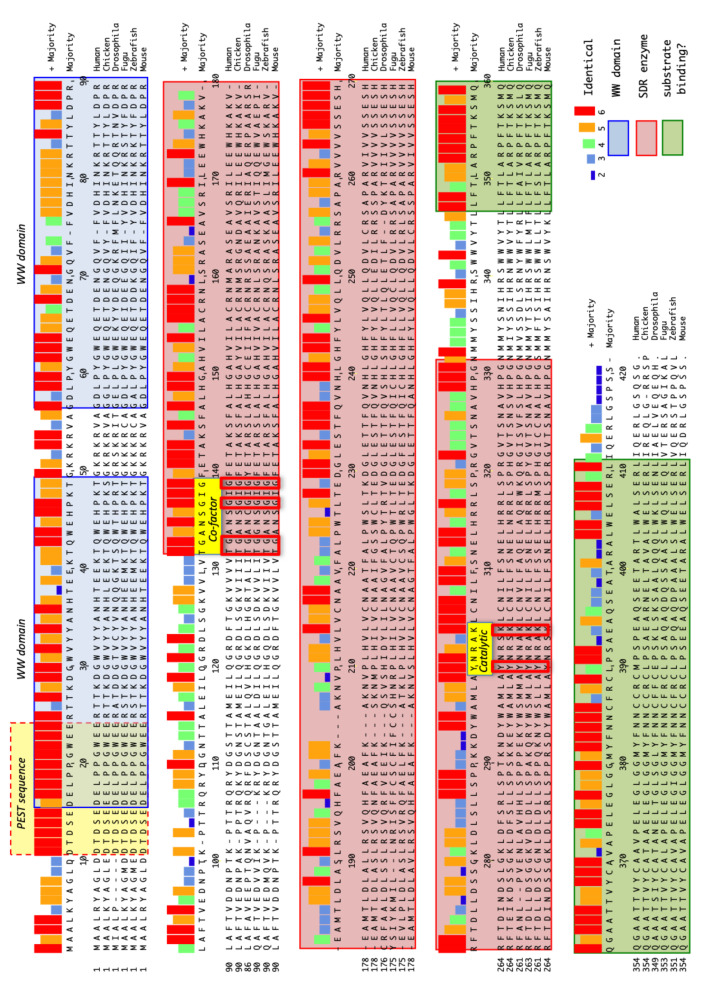
Homology between *WWOX* proteins of different species—Human, Chicken, *Drosophila*, *Fugu*, Zebrafish, Mouse. In addition to WW domains and SDR canonical sequences, the putative substrate specificity sequences are indicated (green shading). Not shown but noted, the *WWOX* orthologue in the evolutionarily distant sea sponge (*Amphimedon queenslandica*) has WW domains and also has substrate binding domain homology. Furthermore, also not shown but noted, the closest SDR family members in the Opisthokonts (*Casaspora owczarzaki*) and *Caenorhabditis elegans* (NP_503155.4 and NP_495501.1) do not have WW domains but do have homology in the putative substrate specificity domain. Highly conserved PEST sequence domain (with 14 of 15 amino acids identical) and the cofactor and catalytic sites that are typical of SDR enzymes are indicated (yellow shading).

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
