# Peer review of "Molecular Biology of the WWOX Gene That Spans Chromosomal Fragile Site FRA16D"

_cells, 2021, doi:10.3390/cells10071637_

Round 1
Reviewer 1 Report
- Generally speaking, the topic of the current draft is not that clear. From the title, I would expect more information on the role of WWOX in the physical and pathological metabolic processes. However, the abstract summarizes the following topics mentioned in detail in the main text: the regular biological role of WWOX using the Drosophila model, relation between WWOX and FRA16D, and WWOX enzyme activity related product. Obviously, from my perspective, the title and the abstract are not focusing on the same topics. So, please either improve the title or reorganize the text.
- The role of WWOX in metabolic disease or the metabolism of cancer disease has been reported these years. To include more recent studies to support the topic is highly suggested.
- If applicable, please provide the full name of the WWOX gene at the right site.
Reviewer 2 Report
1) Within Section 2 ¶ 2 and Section 3 ¶ 4, the authors discuss the long gene length of FHIT and WWOX, their conservation at fragile sites, and the expression of those fragile sites in cancer, therefore, it seems necessary to mention the proposed mechanisms of CFS fragility: infrequency of replication origins (Debatisse lab), replication fork collisions with R loops (Tora lab), and the presence of cis-acting DNA secondary structures in CFSs (Kerem lab). While I understand this is not the focus of this review, a sentence or two provide frame the emerging theories within the CF field seems important to provide to readers. If this review is part of a collection that discusses CFS and these topics are covered in a companion review, please reference it. To further substantiate these sections, please provide references for the statements made within the first 2 sentences of Section 3, ¶ 4.
2) I desperately urge the authors to move Section 5b, explaining Wwox protein domains, earlier in the review—at least before Sections 3 and 4. It is necessary for the readers to understand what is known about normal Wwox protein function in order to better understand the data discussed in Sections 3 and 4. As it is written, readers are formulating a long list of questions about Wwox protein and domains that they carry throughout the review until their questions are answered at the very end. While the review is well-written, the current organization hinders its clarity.
Reviewer 3 Report
In this Review Lee et al. nicely summarize some areas related to WWOX namely: fragile sites, WWOX in cancer and metabolism.
General comment:
Overall the review is fine but should be improved and mistakes corrected in some sections. Some significant papers in each area are reviewed but also a significant number of primary papers and other recent reviews have been overlooked or ignored.
Specific comments:
1) The order or referencing the discovery of WWOX should be chronological. Bednarek et al. Cancer Res 2000 preceded the publication of Ried et al. by several months and the WWOX coding sequence was available in the public domain since 07/12/1999 see GeneBank ID AF212843 (https://www.ncbi.nlm.nih.gov/nuccore/6729684). This group named the gene WWOX. Unfortunately, at that time, Ried et al completely ignored the previous findings and publication from months earlier by Bednarek et al. and named the gene differently. Regardless of any controversy Reference #3 should precede reference #2.
2) The extensive discussion on FHIT, which in itself is hard to follow the way is written, appears completely unnecessary. The reader has no idea what the its 5_’,_5_”-P1,P3-triphosphate hydrolase activity means for example. The parallel discussion of FHIT and WWOX that follows is also puzzling. The fact that FHIT and WWOX are relatively small proteins is completely irrelevant. These are two completely different and unrelated proteins. The topic of the review of Lee et al. is WWOX , the FHIT section should be significantly shorten and preferably completely deleted , it doesn’t add anything to the topic of this special issue on WWOX.
3) Additionally, the focus on the chromosomal region containing WWOX was not necessarily a ‘follow up’ of FRA3B studies as apparently indicated in the text. Multiple groups were studying the region simply because the chromosomal region was of much interest supposedly containing a tumor suppressor gene (LOH etc. ) as indicated later in the text.
4) The comment on ‘patents’ on the gene is completely irrelevant and doesn’t add anything to the topic and should be deleted. We now know that ‘genes’ cannot be patented. What’s the point of this?.
5) Chromosomal instability affecting WWOX and FRA16D l has been recently reviewed by others. It would be great if discussion in this review considers conclusions by others such as https://pubmed.ncbi.nlm.nih.gov/30350478/ , this reference could be included and whether differences in opinion exist, this should be pointed.
6) What do the authors mean by : …”the lack of a relationship between FRA16D associated deletion and another form of deletion (loss-of-heterozygosity, LOH) known to occur at high frequency in certain cancers in the 16q23.2 region”…. ?? . Based on what do the authors conclude there is a ‘lack of relationship’ and these are ‘different forms of deletion’ ? LOH is not ‘another’ form of deletion it is ‘indeed’ a consequence of fragility at FRA16D and it is simply monoallelic and very likely an associated step in the process leading to biallelic deletions in many instances. In many instances in fact compound heterozygous mutations(deletions) have been described (somatic and germinal).
7) WWOX and metabolism section: In agreement with some of the findings by this group in Drosophila, recent TAP-MS studies identified significant enrichment in WWOX protein/complex interactors belonging to metabolic pathways, carbohydrates, and lipids breakdown. Thus, suggesting that WWOX likely plays relevant roles in glycolysis, fatty acid degradation and other pathways that converge primarily in Acetyl-CoA generation as seen in Drosophila and as referred to in such study. Perhaps it would be worth to refer to this work see: https://pubmed.ncbi.nlm.nih.gov/30619736/ further strengthening the point the authors are making.
8) The most important point to clarify. Human WWOX’s SDR catalytic site spans 293YNRSK297 see Fig2a of reference 3 and Fig. 1 of reference 47 of your review, and WWOX protein at position 288 is W not Y see also sequence at https://www.ncbi.nlm.nih.gov/protein/NP_057457.1. Thus, it appears there is a mistake in referring to Y288 as a critical amino acid since it is incorrect for the human sequence and is not the catalytic site. SDR catalytic site should always be YXXXK See also: https://www.ncbi.nlm.nih.gov/Structure/cdd/cddsrv.cgi?uid=187669
Perhaps there is a confusion with the Drosophila sequence? Review carefully the sequence and the statements and Figure 2 clarifying when talking about Y288 to what species you are referring to.
9) Statement: 'The genomic region containing the WWOX gene has been identified as a genetic risk factor in each of these metabolic diseases [36-46].' Ref 47 belongs to this statement and should be included here as well
10) Statement : Homozygous mutations that diminish WWOX function are found in families with recessive spinocerebellar ataxia 12 (SCAR12) [48-50]. The reference describing SCAR12 is none of the references pointed here. Refs 48-50 are describing complete loss of WWOX function in WOREE cases. The only reference describing SCAR12 is Mallaret et al https://pubmed.ncbi.nlm.nih.gov/24369382/ which is also the reference describing the P47T and the G372R mutations and is nowhere to be found. This must be corrected.
11) The mutation in Lde/Lde rats has nothing to do with substrate binding it produces the equivalent to a full knock out by the frameshift (scrambled protein). Also G372R has likely nothing to do with ‘substrate binding’ which is again the catalytic site were the substrate should bind, the functional impact of G372R is unknown.
12) Their speculation on Fig2 of a very large region on the carboxy terminus of the protein for 'substrate binding' makes little to no sense, at least in my view. By definition the ‘substrate binding site’ of an enzyme IS the catalytic site. Additionally the ‘molecule’ that will be impacted by the oxidoreduction properties of WWOX is likely going to be a ‘small molecule’ like for instance an ‘steroid hormone’ like in the case of 17beta dehydrogenases. These are rather small molecules it makes no sense to propose a large ‘substrate binding site’ , again the substrate binding site IS the catalytic site.
Author Response
Reviewers’ comments in blue, responses in black, changes to manuscript in red REVIEWER #1
1. Generally speaking, the topic of the current draft is not that clear. From the title, I would expect more information on the role of WWOX in the physical and pathological metabolic processes. However, the abstract summarizes the following topics mentioned in detail in the main text: the regular biological role of WWOX using the Drosophila model, relation between WWOX and FRA16D, and WWOX enzyme activity related product. Obviously, from my perspective, the title and the abstract are not focusing on the same topics. So, please either improve the title or reorganize the text.
The title has been changed from that assigned by the editor as follows:-
Molecular biology of the WWOX gene that spans chromosomal fragile site FRA16D
2. The role of WWOX in metabolic disease or the metabolism of cancer disease has been reported
these years. To include more recent studies to support the topic is highly suggested.
From their titles the cited manuscripts contain extensive data and discussion of the roles of WWOX in metabolism - we have referenced these manuscripts so that the reader can follow them up for details. We have not attempted a comprehensive review of all that is known about WWWOX as we understand that our review is one of many on the WWOX protein and assume that other reviews will contain more detail on this topic as this is of more focussed interest for other researchers. In addition most publications are reporting the consequences of WWOX role in metabolism rather than its proximal activity - ie its substrate and product in the reaction that it catalyses and how these molecules contribute to metabolism. To our knowledge the substrate(s) and product(s) of WWOX have not yet been identified.
Citations that discuss WWOX and metabolism
24. Abu-Remaileh M, Aqeilan RI. (2015) The tumor suppressor WW domain-containing oxidoreductase modulates cell metabolism Exp Biol Med (Maywood). 240(3), 345-350.
25. O’Keefe, L.V., Colella, A., Dayan, S., Chen, Q., Choo, A., Jacob, R., Price, G., Venter, D. and Richards, R.I (2011) Drosophila orthologue of WWOX, the chromosomal fragile site FRA16D tumour suppressor gene, functions in aerobic metabolism and regulates reactive oxygen species Human Molecular Genetics 20, 497–509.
26. Dayan, S., O’Keefe, L.V., Choo, A., and Richards, R.I (2013) Common Chromosomal Fragile Site FRA16D Tumour Suppressor WWOX Gene Expression and Metabolic Reprogramming in Cells. Genes, Chromosomes and Cancer 52, 823-831.
27. O’Keefe, L.V., Lee, C.S., Choo, A., and Richards, R.I. (2015) Tumor suppressor WWOX contributes to the elimination of tumorigenic cells. PLOS One 10(8), e0136356.
28. Choo, A., O’Keefe, L.V., Lee, C.S., Gregory, S.L., Shaukat, Z., Colella, A. Lee, K., Denton, D. and Richards, R.I. (2015) Tumour suppressor WWOX moderates the mitochondrial respiratory complex. Genes, Chromosomes and Cancer 54, 745-761.
32. Abu-Remaileh M, Aqeilan RI. (2014) Tumor suppressor WWOX regulates glucose metabolism via HIF1alpha modulation. Cell Death Differ. 21(11), 1805-1814.
33. Abu-Remaileh M, Seewaldt VL, Aqeilan RI. (2014) WWOX loss activates aerobic glycolysis. Mol Cell Oncol. 2(2), e965640.
34. Abu-Remaileh M, Abu-Remaileh M, Akkawi R, Knani I, Udi S, Pacold ME, Tam J, Aqeilan RI. (2019) WWOX somatic ablation in skeletal muscles alters glucose metabolism. Mol Metab. 22, 132-140.
35. Iatan I, et al., (2014) The WWOX gene modulates high-density lipoprotein
and lipid metabolism. Circ Cardiovasc Genet. 7(4), 491-504.
3. If applicable, please provide the full name of the WWOX gene at the right site.
It is not clear what the reviewer is asking for here, however we have added :-
WWOX (WW-containing OXidoreducase) and FOR (Fragile site FRA16D Oxido-Reductase) in the hope that these fulfil this reviewers need.
REVIEWER #2
1) Within Section 2 ¶ 2 and Section 3 ¶ 4, the authors discuss the long gene length of FHIT and WWOX, their conservation at fragile sites, and the expression of those fragile sites in cancer, therefore, it seems necessary to mention the proposed mechanisms of CFS fragility: infrequency of replication origins (Debatisse lab), replication fork collisions with R loops (Tora lab), and the presence of cis-acting DNA secondary structures in CFSs (Kerem lab). While I understand this is not the focus of this review, a sentence or two provide frame the emerging theories within the CF field seems important to provide to readers. If this review is part of a collection that discusses CFS and these topics are covered in a companion review, please reference it. To further substantiate these sections, please provide references for the statements made within the first 2 sentences of Section 3, ¶ 4.
We respectfully disagree.
We understand that this review is meant to be about WWOX not fragile sites. We have included discussion of fragile sites in so much as the WWOX gene (in both human and mouse) contains one and that the genes very large size is conserved in evolution - suggesting that there is some (as yet unknown) biological relationship between the WWOX gene and chromosome fragility. We have not attempted to review the current literature on chromosomal fragile sites and their genes, indeed reviewer #3 is critical of the amount of discussion that we have included on the FHIT gene that spans the FRA3B fragile site, even though this is pertains to whether there is a general relationship between fragile sites and the genes that span them (including WWOX).
We have amended section 1 and added the following and the Glover review reference :-
A correlation between cancer cell DNA instability and chromosomal fragile sites had been noted long ago [8], although the notion that such a relationship was causal had been greeted with some scepticism [9,10]. A relationship of some sort received a boost of interest with the finding that the most readily observed common chromosomal fragile site FRA3B, was located within a region on human chromosome 3 that exhibited DNA instability in cancer [10]. Furthermore, the FHIT gene was found to span the FRA3B common chromosomal fragile site and aberrant transcripts of the FHIT gene were found in cancer cells [11]. A functional role for FHIT as a tumour suppressor however also turned out to be controversial with conflicting data regarding the contribution of FHIT to cancer [12, 13]. In one study using neoplastic cells that had FRA3B deletions and therefore were deficient in FHIT protein, !replacement” with stable, over-expressed FHIT protein did not alter in vitro or in vivo properties of these cells [14]. In another study [15], replacement of FHIT protein in cancer cells suppressed their tumorigenicity. To further add to the mystery ‘enzyme inactive’ mutant FHIT was just as effective as the normal active FHIT at suppressing tumourigenicity [15] implying that its 5’,5”-P1,P3-triphosphate hydrolase activity is not required for all of its functions. The review by Glover et al., (2017) details recently reported mechanisms of the cytogenetic expression of common chromosomal fragile sites and some of the controversy around whether or not WWOX and FHIT are actually tumour suppressor genes.
Glover, T.W., Wilson, T.E., Arlt, M.F. (2017) Fragile Sites in Cancer: More Than Meets the Eye. Nat Rev Cancer. 17(8): 489–501.
As this review contains more details on the recently reported mechanisms of fragile site expression and some of the controversy around whether or not WWOX and FHIT are actually tumour suppressor genes.
2) I desperately urge the authors to move Section 5b, explaining Wwox protein domains, earlier in the review—at least before Sections 3 and 4. It is necessary for the readers to understand what is known about normal Wwox protein function in order to better understand the data discussed in Sections 3 and 4. As it is written, readers are formulating a long list of questions about Wwox protein and domains that they carry throughout the review until their questions are answered at the very end. While the review is well-written, the current organisation hinders its clarity.
The substrate and product of the WWOX enzyme activity are not known - the reader will get to the end of our review and not have their questions answered because until this is known all of the data that accumulates about the roles of WWOX will be about the downstream consequences of the reaction that it catalyses. A major aim of our review is to point to the fact that despite more than twenty years of research on the WWOX protein the enzyme reaction that it catalyses is still unknown and needs to be discovered.
The following ‘spoiler alert’ statement has therefore been added at the beginning of section 3
Despite more than twenty years of research on the WWOX protein, the substrate and product of the enzyme reaction that it catalyses are yet to be discovered.
REVIEWER #3
1) The order or referencing the discovery of WWOX should be chronological. Bednarek et al. Cancer Res 2000 preceded the publication of Ried et al. by several months and the WWOX coding sequence was available in the public domain since 07/12/1999 see GeneBank ID AF212843 (https://www.ncbi.nlm.nih.gov/nuccore/ 6729684). This group named the gene WWOX. Unfortunately, at that time, Ried et al completely ignored the previous findings and publication from months earlier by Bednarek et al. and named the gene differently. Regardless of any controversy Reference #3 should precede reference #2.
The first report of the gene that would become known as WWOX was by Paige et al (2000) and was an already known expressed sequence tag site for the C-terminal end of an otherwise unknown member of the oxido-reductase protein family HHCMA56 (GenBank accession number U13395) submitted to GenBank in 1994 by Gmerek,R.E. and Medford,J.I. TITLE: “The complete sequence of a human hippocampus gene (HHCMA56) shows homology to developmental genes from Arabidopsis and Brassica napus”.
Paige et al excluded HHCMA56 (also known as D16S432E) from contention on the basis that “We screened DNA from our YAC contig and the deletion-containing cell lines for all six genes by PCR. All of the candidate genes were found to be present in all of the cell lines and therefore must lie outside the deleted regions (data not shown). In addition, all of the loci were found to lie outside of the YAC contig, with the exception of HHCMA56, which is contained within YAC clone 972D3 and therefore lies several hundred kilobases distal of the PAC contig and the minimally deleted region.”
The massive length of the gene and the fact that the minimally deleted region is intronic therefore delayed its definitive identification as that which spans the FRA16D fragile site. In addition to the name WWOX this gene therefore now has a list of alternate names (FOR; WOX1; DEE28; EIEE28; FRA16D; SCAR12; HHCMA56; PRO0128; SDR41C1; D16S432E). The Human Gene Nomenclature Committee chose the name WWOX and therefore we and other laboratories have been referring to as such in an effort to minimise any confusion in the field.
The order of the references has been amended such that Bednarek et al (2000) now appears before Ried et al. (2000) as requested by Reviewer #3. The Paige et al (2000) reference has been added along with its mapping of HHCMA56 to the region and discussion of the basis of its exclusion from contention and relevance to the properties of the gene, as follows :-
Indeed the experiments leading to the identification of the WWOX gene are noteworthy as they reveal a number of significant (and unexpected) characteristics, which have an impact on the encoded protein and its function. Some of these are yet to be explained.
A major reason for interest in this region of the genome was based upon the consequences Knudsen hypothesis (ref) - that inherited cases of cancer were higher than sporadic cases because the latter required two somatic mutations to a tumour suppressor gene while the familial cases only needed one. One of the forms of second mutation that validated Knudsen’s hypothesis was loss- of-heterozygosity and so the search for regions of the genome that exhibited loss-of-heterozygosity in cancer was thought to be a means of tracking down novel tumour suppressor genes. The FRA16D region had been found to be within overlapping regions of loss-of-heterozygosity in breast (Chen et al., 1996) and prostate (Latil et al.,1997) cancers suggesting the presence of a tumour suppressor. The presence of FRA16D, the second most readily observed common chromosomal fragile site in the human genome, contributed to speculation of a causal relationship between DNA fragility and instability and the presence of a tumour suppressor gene. Indeed both Mangelsdorf et al (2000) and Paige et al., (2000) identified homozygous deletions within several cancer cell lines that coincided with the location of FRA16D.
The gene that has come to be known as WWOX (WW-containing Oxidoreducase, Bednarek et al., 2000) was first located in the FRA16D region by Paige et al., (2000) as HHCMA56, an oxidoreductase encoding sequence that had been deposited in GenBank in 1994 by Gmerek, R.E. and Medford, J.I. Paige et al. (2000) had excluded HHCMA56 from contention on the basis of a PCR (D16S432E) that located (the final exon) of this gene hundreds of kilobases in distance from the “minimal homozygously deleted region in cancer cells” identified by Mangelsdorf et al., (2000) and Paige et al., (2000). Oxidoreductases were not amongst known tumour suppressors at the time and HHCMA56 therefore appeared to have unlikely credentials as such, although some members of this protein family have since been found to have modifying roles in cancer (Deisenroth et al., 2010; Li et al., 2019). The FRA16D minimal deletion region was in due course sequenced by Ried et al (GenBank accession number AF217490) as it was expected to contain within it one or more exons of a tumour suppressor gene. Instead the gene responsible for HHMCA56 was found to be huge and indeed span FRA16D. With hindsight such a possibility might have been considered given that the huge FHIT gene spanned the FRA3B fragile site.
HHCMA56 was a partial cDNA sequence from one of a number of alternatively spliced RNA transcripts. D16S432E is located in its unique 3’ exon (corresponding to exon 9 of WWOX). Exon 8 of WWOX is shared between two alternatively spliced transcripts (named FOR I and FORII by Ried et al, 2000). The common exon 8 sequences were found at the very beginning of the AF217490 sequence (indeed prior to the minimally deleted region) and the alternative exon 9 from the FOR I transcript at the other end. Finnis et al (2005) subsequently found that some homozygous deletions in cancer cells are indeed only intronic, which appears at odds with such deletions knocking out a tumour suppressor gene. This intron is 260kb in length in the minor FOR I transcript and a massive 780kb in length in the major FORII (WWOX) transcript. Furthermore both transcripts share intron 5 that is also a massive 222570 bases in length. The cancer cells with these homozygous deletions are still able to exhibit cytogenetic expression of the fragile site indicating that sequences outside of the region of DNA deletion are responsible for fragility and presumably DNA instability.
The gene was subsequently named WWOX (WW domain containing Oxido-reductase) by Bednarek et al., (2000) while Ried et al., (2000) had given the gene the name of FOR (Fragile site FRA16D Oxido-Reductase). Chang et al., (2001) identified the gene by virtue of its induction by hyaluronidase and named it WOX1. The WWOX gene has therefore accumulated alternate names (FOR; WOX1; DEE28; EIEE28; FRA16D; SCAR12; HHCMA56; PRO0128; SDR41C1; D16S432E).
The WWOX gene is of remarkable length for a protein of only 414 amino acids. Its primary transcript is over 1.1Mb in length of which >98% is intron. For comparison another member of the SDR family to which WWOX belongs is hydroxysteroid 17-beta dehydrogenase 1 encoded by the gene HSD17B1. This protein of 328 amino acids is translated from a mature mRNA of 1,274 nucleotides having been spliced from a primary gene transcript of 2,292 nucleotides in length. Not only does the human WWOX gene have vastly larger introns that another member of the same enzyme encoding gene family but this extreme length of introns has been conserved through evolution, even in organisms that typically have short introns, i.e. Fugu and Drosophila (Figure 1). The parallels with the FHIT gene that spans the FRA3B common chromosomal fragile site are striking and of relevance to further properties of WWOX as discussed later in this review.
Steady-state protein abundance is determined by four rates: transcription, translation, mRNA decay and protein decay (Hausser et al., 2019). The ability of a gene to produce a protein product is determined by transcription which takes time. Transcription rates vary widely however it is safe to assume that a primary transcript of the WWOX gene takes several of not many hours to complete and is significant in relation to the time necessary for the cell cycle in dividing cells. FRA16D associated intronic deletions might be expected to hasten the process however introns and their splicing can enhance gene expression (Shaul, 2017). Adding to the enigma WWOX primary transcripts undergo alternative splicing with only one form encoding the full-length protein. Typically transcripts alternative to the full length are subject to non-sense mediated decay and contribute to a reduction in the steady-state level of mRNA for the full-length protein. Driouch et al., (2002) however report a substantially elevated level an alternatively spliced transcript (designated FORIII by Ried et al., 2000) in ~50% of breast cancer tissues and cell lines. This perturbed splicing occurs in the absence of detectable DNA deletions within the WWOX gene, further contributing to the enigma. Its relevance to cancer cell biology is as yet unknown.
The enormous length of the WWOX gene and its alternative splicing would appear to be two of the contributing factors to WWOX protein having a low steady-state level. Indeed Drosophila go one step further with the presence of an intron in the 3’ untranslated region (UTR) of WWOX transcripts (see Wwox-RB versus Wwox-RA transcripts in FlyBase (https://flybase.org/ reports/FBtr0343384). Bicknell et al., report that mRNAs with 3’UTR introns are subject to non-
sense mediated decay after only a single round of translation, indicating another mechanism for keeping WWOX levels low. Whether human WWOX RNA transcripts also have such 3’UTR introns may warrant further investigation as according to Bicknell et al., it is often assumed that such sequences are non-functional.
The authors thank Reviewer #3 for pointing out the need to recount these matters that have long been in the literature but have been given relatively little attention in further efforts to understand not only the relationship between common chromosomal fragile sites and the genes that span them but also their contribution to biology. It is likely that understanding the normal role of WWOX and its relationship to chromosomal fragility are necessary in order to discover how the perturbation of these normal roles results in disease.
For Reviewer #3’s information
Publication
Mangelsdorf et al., Paige et al., Bednarek et al., Ried et al.,
date of publication
March 15, 2000 March 15, 2000 April 15, 2000 April 27, 2000
The AF212843 sequence was not public at the time of submission of Ried et al., (2000). i.e. It definitely did not show up on BLAST searches up to that time.
GeneBank policy states :-
Confidentiality
Some authors are concerned that the appearance of their data in GenBank prior to publication will compromise their work. GenBank will, upon request, withhold release of new submissions for a specified period of time. However, if the accession number or sequence data appears in print or online prior to the specified date, your sequence will be released. In order to prevent the delay in the appearance of published sequence data, we urge authors to inform us of the appearance of the published data. As soon as it is available, please send the full publication data--all authors, title, journal, volume, pages and date--to the following address: update@ncbi.nlm.nih.gov
The publications of Mangelsdorf et al and Paige et al cited one another because the authors were in communication during the publication process. Bednarek et al cites neither of these manuscripts presumably because it was submitted before they were published - as was the case for Ried et al (submitted on 13th March) and the Bednarek et al publication. These details are supplied to Reviewer #3 simply in order to set the record straight. It is difficult to know how much (if any) of this publication information is of interest to anyone else in the field so has not been included in the revised text.
2) The extensive discussion on FHIT, which in itself is hard to follow the way is written, appears completely unnecessary. The reader has no idea what the its 5’,5”-P1,P3-triphosphate hydrolase activity means for example. The parallel discussion of FHIT and WWOX that follows is also puzzling. The fact that FHIT and WWOX are relatively small proteins is completely irrelevant. These are two completely different and unrelated proteins. The topic of the review of Lee et al. is WWOX , the FHIT section should be significantly shorten and preferably completely deleted , it doesn’t add anything to the topic of this special issue on WWOX.
We respectfully disagree.
FHIT is a gene that was identified on the basis of it spanning the most frequently observed common chromosomal fragile site, FRA3B, and making a contribution to cancer - presumably as a tumour suppressor. These are important attributes shared with WWOX and FRA16D (the second most frequently observed human chromosomal fragile site). The relationship between common chromosomal fragile sites and DNA instability in cancer has been of great interest for a long period of time (Yunis and Soreng, 1984). The mouse orthologs of the FHIT and WWOX genes also span common chromosomal fragile sites. This conservation during evolution strongly indicates that there is a normal biological relationship between the chromosomal predisposition to fragility and the particular genes in which they are located. Therefore the function of FHIT is of interest to at least some in the research community studying WWOX. These proteins could for example be contributing to a common biological pathway and / or process. The lessons learned from research in to the role or otherwise of FHIT in cancer are relevant to the role of WWOX in cancer - both have been described as ‘non-classical tumour suppressors’ in an effort to accommodate apparently conflicting or at least enigmatic data.
Others working on WWOX cite and discuss FHIT literature because it is relevant eg see former reference 47. Aldaz, C.M., et al., (2014) WWOX at the crossroads of cancer, metabolic syndrome related traits and CNS pathologies. Biochim Biophys Acta. 1846, 188-200.
The title of the manuscript has been changed from that which was given to us by the editor to more closely reflect our interest in the WWOX gene and its association with one of the most frequently observed human common chromosomal fragile sites.
New TITLE:
Molecular biology of the WWOX gene that spans chromosomal fragile site FRA16D
3) Additionally, the focus on the chromosomal region containing WWOX was not necessarily a ‘follow up’ of FRA3B studies as apparently indicated in the text. Multiple groups were studying the region simply because the chromosomal region was of much interest supposedly containing a tumor suppressor gene (LOH etc. ) as indicated later in the text.
As stated above, the relationship between common chromosomal fragile sites and DNA instability in cancer predates research aimed at linking DNA instability on chromosome 16 to cancer (see Yunis and Soreng, 1984). The findings at FRA3B informed the need for analysis by some other laboratories of the next most frequently observed common fragile site, FRA16D.
4) The comment on ‘patents’ on the gene is completely irrelevant and doesn’t add anything to the topic and should be deleted. We now know that ‘genes’ cannot be patented. What’s the point of this?.
Discussion of patents is deleted and replaced with the response to #1.
5) Chromosomal instability affecting WWOX and FRA16D l has been recently reviewed by others. It would be great if discussion in this review considers conclusions by others such as https:// pubmed.ncbi.nlm.nih.gov/30350478/ , this reference could be included and whether differences in opinion exist, this should be pointed.
The reference
6) What do the authors mean by : ...”the lack of a relationship between FRA16D associated deletion and another form of deletion (loss-of-heterozygosity, LOH) known to occur at high frequency in certain cancers in the 16q23.2 region”.... ?? . Based on what do the authors conclude there is a ‘lack of relationship’ and these are ‘different forms of deletion’ ? LOH is not ‘another’ form of deletion it is ‘indeed’ a consequence of fragility at FRA16D and it is simply monoallelic and very likely an associated step in the process leading to biallelic deletions in many instances. In many instances in fact compound heterozygous mutations(deletions) have been described (somatic and germinal).
The section entitled “Relationship between FRA16D homozygous deletions and 1623.2 LOH” on page 1344 of the Finnis et al (2005) manuscript details the experimental basis for the conclusion that the homozygous deletion events observed in cancer cell lines in this manuscript are distinct from the loss-of-heterozygosity observed by others in breast (Chen et al., 1997) and prostate (Latil et al., 1997) cancer. In brief, the polymorphic genetic markers D16S518 and D16S504 that define the boundaries of the loss-of-heterozygosity regions identified in cancer (by Chen et al 1997 and Latil et al., 1997), were found by Finnis et al (2005) to be heterozygous in all of the cancer cell lines that exhibited homozygous deletion. This indicates that the homozygous deletions observed in the cancer cell lines under investigation (ie AGS, HCT116, CO-115, KM12C and KM12SM) are not able to be the boundaries of any extensive loss-of-heterozygosity, at least in these particular cancer cell lines. The KM12 lines had a common origin and were found to retain identical homozygous deletions at FRA16D yet exhibited different DNA instabilities elsewhere (including a chromosomal translocation) indicating that the fragile site DNA instability is not always tied to other instances of DNA instability that presumably have different causes.
Hussain et al., (2019) WWOX, the FRA16D gene: A target of and a contributor to genomic
instability. Genes Chromosomes Cancer 58(5):324-338. doi: 10.1002/gcc.22693.
is now included.
The following has therefore been added to the text (and cited references added)
The relationship between FRA16D homozygous deletion and the loss-of-heterozygosity is
particularly noteworthy. Finnis et al (2005) detail the experimental basis for the conclusion that the FRA16D associated homozygous deletion events observed in cancer cell lines in this manuscript are distinct from the loss-of-heterozygosity observed by others in breast (Chen et al., 1996) and prostate (Latil et al.,1997) cancers. In brief, the polymorphic genetic markers D16S518 and D16S504 that define the boundaries of the loss-of-heterozygosity regions identified in cancer (by Chen et al., 1996 and Latil et al., 1997), were found by Finnis et al (2005) to be heterozygous in all of the cancer cell lines that exhibited FRA16D homozygous deletion. This indicates that the homozygous deletions observed at least in the particular cancer cells under investigation (ie AGS, HCT116, CO-115, KM12C and KM12SM) are not able to be the boundaries of any loss-of- heterozygosity that may have occurred in the vicinity in these cells. Whether this nexus is broken in other instances is yet to be determined. It is perhaps noteworthy that the KM12 lines had a common origin (being derived from the primary, KM12C and metastasis, KM12SM of the same cancer) and had identical homozygous deletions at FRA16D yet exhibited different DNA instabilities elsewhere (including a chromosomal translocation). Fragile site DNA instability is therefore not always tied to other instances of DNA instability that presumably have different causes.
7) WWOX and metabolism section: In agreement with some of the findings by this group in Drosophila, recent TAP-MS studies identified significant enrichment in WWOX protein/complex interactors belonging to metabolic pathways, carbohydrates, and lipids breakdown. Thus, suggesting that WWOX likely plays relevant roles in glycolysis, fatty acid degradation and other pathways that converge primarily in Acetyl-CoA generation as seen in Drosophila and as referred to in such study. Perhaps it would be worth to refer to this work see: https:// pubmed.ncbi.nlm.nih.gov/30619736/ further strengthening the point the authors are making.
8) The most important point to clarify. Human WWOX’s SDR catalytic site spans 293YNRSK297 see Fig2a of reference 3 and Fig. 1 of reference 47 of your review, and WWOX protein at position 288 is W not Y see also sequence at https://www.ncbi.nlm.nih.gov/protein/NP_057457.1. Thus, it appears there is a mistake in referring to Y288 as a critical amino acid since it is incorrect for the human sequence and is not the catalytic site. SDR catalytic site should always be YXXXK See also: https://www.ncbi.nlm.nih.gov/Structure/cdd/cddsrv.cgi? uid=187669
Perhaps there is a confusion with the Drosophila sequence? Review carefully the sequence and the statements and Figure 2 clarifying when talking about Y288 to what species you are referring to.
This text refers to the Drosophila WWOX protein and experiments in which amino acid position at 288 is the tyrosine reside at the start of the catalytic site motif and therefore the appropriate amino acid to alter (to phenylalanine) in order to assess the need for catalytic activity for the contribution of WWOX to metabolism (as is pointed out in the Choo et al., manuscript).
The revised manuscript has been amended as follows:-
The orthologous tyrosine amino acid is position 293 in human WWOX. The WWOX proteins of different species vary in length due to the presence / absence of additional amino acids.
9) Statement: 'The genomic region containing the WWOX gene has been identified as a genetic risk factor in each of these metabolic diseases [36-46].' Ref 47 belongs to this statement and should be included here as well
Reference #47 is a review that contains no new data on this topic that is novel to the primary research publications that it cites. Never-the-less as it has been cited elsewhere and it has been added again here simply in order to comply with the reviewers wishes.
10) Statement : Homozygous mutations that diminish WWOX function are found in families with recessive spinocerebellar ataxia 12 (SCAR12) [48-50]. The reference describing SCAR12 is none
Hussain et al, (2018) is now also cited in the revised manuscript as follows:-
Pathway analysis of WWOX interactors by Hussain et al., (2018) identified a significant enrichment
of metabolic pathways associated with proteins, carbohydrates, and lipids breakdown.
of the references pointed here. Refs 48-50 are describing complete loss of WWOX function in WOREE cases. The only reference describing SCAR12 is Mallaret et al https:// pubmed.ncbi.nlm.nih.gov/24369382/ which is also the reference describing the P47T and the G372R mutations and is nowhere to be found. This must be corrected.
The Mallaret et al (2014) reference has been added. The data from it was in the review by Piard et al (2019) that is also cited. The numbers of the other relevant references have also been amended as they were incorrect in the original submission.
11) The mutation in Lde/Lde rats has nothing to do with substrate binding it produces the equivalent to a full knock out by the frameshift (scrambled protein). Also G372R has likely nothing to do with ‘substrate binding’ which is again the catalytic site were the substrate should bind, the functional impact of G372R is unknown.
We agree with the reviewer - the lde mutation does indeed appear to have an impact on protein stability that could account for the observed phenotype as a complete loss-of-function. Whilst the very low endogenous abundance of WWOX protein renders its detection problematic the western blot of Suzuki et al (2009) indicates at least a reduction in WWOX levels.
The Suzuki et al (2009) reference has therefore been removed. The Mallaret et al. (2014), reference is now included (as mentioned above) with the comment that the G372R mutation (which is located within the highly conserved C-terminal region) has very similar clinical consequences to that of P47T (located in the first WW domain).
12) Their speculation on Fig2 of a very large region on the carboxy terminus of the protein for 'substrate binding' makes little to no sense, at least in my view. By definition the ‘substrate binding site’ of an enzyme IS the catalytic site. Additionally the ‘molecule’ that will be impacted by the oxidoreduction properties of WWOX is likely going to be a ‘small molecule’ like for instance an ‘steroid hormone’ like in the case of 17beta dehydrogenases. These are rather small molecules it makes no sense to propose a large ‘substrate binding site’ , again the substrate binding site IS the catalytic site.
We defer to those working in the SDR enzyme field and have included the reference Kavanagh et al “The SDR superfamily: functional and structural diversity within a family of metabolic and regulatory enzymes” is now included and cited with the following quotation:-
“The common mechanism is an underlying hydride and proton transfer involving the nicotinamide and typically an active site tyrosine residue, whereas substrate specificity is determined by a variable C-terminal segment.”
The observed C-terminal regions of conservation during evolution do not exhibit any detectable homology to known protein motifs. They look to be a unique property of the WWOX protein and therefore somehow related to its unique biological function.
It is therefore reasonable to put forward the idea that sequences outside of the immediate catalytic motif (and C-terminal to it) have a role to play in contributing to conformation of the catalytic site and / or the access of molecules to that catalytic site and therefore the specificity of the enzyme. Whatever the biological basis of the very high level of conservation during evolution in the C- terminal region of WWOX, it warrants drawing attention to and discussion in the hope that this stimulates further research and the discovery of its actual biological basis. One would hope that this purpose of review articles is appreciated and respected.
We have added to the discussion of this section as follows:-
Whilst conserved during evolution these sequences do not exhibit any detectable homology to known protein motifs. They look to be a unique property of the WWOX protein and therefore somehow related to its unique biological function.
Kavanagh et al in their review entitled “The SDR superfamily: functional and structural diversity within a family of metabolic and regulatory enzymes” state the following “The common mechanism is an underlying hydride and proton transfer involving the nicotinamide and typically an active site tyrosine residue, whereas substrate specificity is determined by a variable C-terminal segment.” Whatever the biological basis of the very high level of conservation during evolution in the C-terminal region of WWOX, it is reasonable to speculate that sequences outside of the
immediate catalytic motif (and C-terminal to it) have a role to play in contributing to conformation of the catalytic site and / or the access of molecules to that catalytic site and therefore the specificity of the enzyme.
Of relevance, homozygous mutations that diminish WWOX function are found in families with recessive spinocerebellar ataxia 12 (SCAR12) [48-50, Mallaret, 2014]. The G372R mutation located within the putative substrate specificity region indicates that this C-terminal region is vital for WWOX function, having the same clinical consequences as that of the P47T mutation located in the first WW domain of WWOX (Mallaret). Given that these conserved C-terminal sequences do contribute to WWOX substrate-specificity then they are potential targets in the identification of WWOX-based therapeutics.

Round 2
Reviewer 3 Report
The authors did a great job with the revision of this manuscript improving significantly from the previous version. Now this new version is far more organized, sections and statements make more sense under adequate subtitles, e.g. the FHIT/FRA3B discussion which now in the proper context is adequate. The change in the title is also a positive modification. The manuscript will be acceptable for publication after minor modifications as indicated below
Specific points:
1) Regarding item #1 of the previous review and response provided by the authors: The authors provide a very long (exhausting to read) explanation of honestly little interest to readers, it was even hard to follow to this reviewer that is aware of the topic.
a) The statement: ‘The gene that has come to be known as WWOX was first located in the FRA16D region by Paige et al as HHCMA56...’. It is misleading and should be modified. Paige at al. described and accurate physical map of the region containing FRA16D/WWOX but by no means cloned the gene WWOX. Indeed, this region contains HHCAM56 as also indicated by Bednarek et al in their paper. But Paige et al. also noted numerous other ESTs in the region and did not clone any gene. HHCAM56 is an EST (not a gene) with extensive mismatches, large gaps and differences compared to the final real transcript and it is just not the complete accurate WWOX sequence it is only a portion of it.
b) D16S432E is not a ‘PCR’ is an STS marker.
c) For the sake of making the reading easier to the uninformed reader this section can be significantly shorten, as is now with all due respect is a pain to read and follow.
d) For the author’s information the dates provided of publication are simply ‘wrong’ as per each corresponding links and pdfs
Bednarek et al. https://cancerres.aacrjournals.org/content/60/8/2140.article-info
Received January 11, 2000
Accepted March 3, 2000
Published first April 1, 2000
Ried et al. https://academic.oup.com/hmg/article/9/11/1651/607056
Received 13 March 2000.
Revised and Accepted 27 April 2000.
Published: 01 July 2000
2) Now the discussion of FHIT is in context and makes sense not like in the previous disorganized version.
3) On the issue of the apparent lack of relationship between LOH and FRA16D homozygous deletion, it is perhaps at least partially explainable by recent observations by Hadi et al., 2020, Cell 183, 197–210 entitled ‘Distinct Classes of Complex Structural Variation Uncovered across Thousands of Cancer Genome Graphs’. Using a novel computation approach these authors classified complex structural variations and one of the classes the authors identified are called Rigma which are chasms of low junction copy number deletions at late replicating fragile sites arising early in tumor evolution, where they precisely point and discuss FHIT and WWOX as examples of Rigma in GI cancers (see Discussion). I encourage the authors to take a look at the study and see whether is useful.